# Nitrile-Specific Protein NSP2 and Its Interacting Protein MPK3 Synergistically Regulate Plant Disease Resistance in *Arabidopsis*

**DOI:** 10.3390/plants12152857

**Published:** 2023-08-03

**Authors:** Tingting Zhai, Jun Teng, Xintong Fan, Shaowei Yu, Chen Wang, Xingqi Guo, Wei Yang, Shuxin Zhang

**Affiliations:** 1National Key Laboratory of Wheat Improvement, College of Life Science, Shandong Agricultural University, Tai’an 271018, China; zhaiting0520@163.com (T.Z.); fxt0928@163.com (X.F.); 123yushaowei@163.com (S.Y.); cwang@sdau.edu.cn (C.W.); 2College of Animal Science and Technology, Shandong Agricultural University, Tai’an 271018, China; tengjun0520@163.com; 3School of Chemical and Biological Engineering, Qilu Institute of Technology, Jinan 271018, China; xqguo@sdau.edu.cn

**Keywords:** *NSP2*, *MPK3*, *Pst* DC3000, phosphorylation, disease resistance

## Abstract

Glucosinolates and their degradation products have a wide range of actions and are important components of plant defense. *NSP2* (nitrile-specific protein 2) is a key regulator in the breakdown process of glucosinolates. However, the precise function of *NSP2* in plant disease resistance beyond its role in glucosinolate degradation is still unclear. In this study, we discovered that *NSP2* which was induced by *Pst* DC3000, influenced *PR* genes expression and reactive oxygen burst. Additionally, omics analysis revealed that NSP2 was engaged in plant-pathogen interaction and several hormone signal transduction pathways. Furthermore, immunoprecipitation-tandem mass spectrometry analysis (IP-MS), bimolecular fluorescence complementation (BiFC), and co-immunoprecipitation demonstrated that NSP2 interacts with MPK3. Genetic analysis shows that *NSP2* may be a function downstream of *MPK3*. Upon pathogen inoculation, NSP2 protein levels increase while MPK3 protein levels decrease. Moreover, the level of phosphorylated NSP2 decreases. Taken together, this study sheds light on a new mode of synergistic action between NSP2 and MPK3 in the disease resistance process.

## 1. Introduction

Glucosinolates are nitrogen and sulfur-containing secondary metabolites found in Brassicales. Intact glucosinolates are thought to be physiologically inactive, whereas breakdown products play a critical role in plant defense against herbivores and pathogens [1,2,3,4,5,6]. Nitrile-specific proteins (NSPs) are a kind of particular protein involved in glucosinolate catabolism [7]. There are several mechanisms for the degradation of glucosinolates, and their breakdown into distinct products is influenced by a variety of circumstances, including the presence or lack of certain functional genes, such as ESP and NSP [8]. In *Arabidopsis thaliana*, there are five members of the *NSP* family, *NSP1*–*NSP5*, whose biological activities are poorly understood [7,9]. Except for *NSP5*, other members of the *NSP* family gene have 1–2 Jacalin structures in the N-terminus, the proteins containing this domain can respond to biological stresses like powdery mildew and gibberella infection, or abiotic stresses like cold injury, high salt, and drought, or hormones like SA, JA, and ABA [10,11]. This suggests that NSP family proteins may be involved in plant stress response. Furthermore, all NSP family proteins include four repeated kelch domains, which are hypothesized to be involved in protein interaction [12].

The mitogen-activated protein kinase (MAPK) cascade pathway is found in all species and is a highly conserved signaling mechanism. This route is essential for plant growth and development, as well as reaction to environmental stress. Activated MAPK serves three important physiological functions: (1) Phosphorylation of cytoskeletal proteins, resulting in cytoskeletal changes; (2) activation of other protein kinases to induce stress signal transmission; (3) translocation into the nucleus, binding to transcription factors, and activating them via phosphorylation, thereby regulating target gene expression. Numerous studies have highlighted a tight relationship between the MAPK signal and plant responses to abiotic stress, indicating that it plays an important role in enhancing plant stress resistance [13,14,15,16,17]. In the context of plant immunity, two MAPK cascades have been extensively studied. Both the MAP3K3/5-MKK4/5-MPK3/6 and MEKK1-MKK1/2-MPK4 cascades are rapidly and transiently activated upon perception of pathogen-associated molecular patterns (PAMPs), leading to gene reprogramming critical for mounting effective pathogen-associated molecular pattern-triggered immunity (PTI) responses [18,19,20,21].

The mechanisms of plant resistance and defense against pathogens mediated by MPK3 and MPK6 involve the regulation of transcriptional activation of defense genes, synthesis of plant antitoxins, cell-wall thickening, hypersensitivity, stomatal closure, endogenous hormone production, and reactive oxygen species (ROS) outbreak [22,23,24,25,26]. For example, HopAI1, a phosphothreonine lyase, can directly interact with and inactivate MPK3 and MPK6, thereby inhibiting gene expression, oxidative burst, and corpus callosum deposition induced by the activator flg22, resulting in increased susceptibility of plants to diseases [27]. In addition, MPK3 and MPK6 can regulate camalexin biosynthesis by phosphorylation of WRKY33 [28]. During the synthesis of camalexin, the indole glucosinolate production also increased, and participated in the immune process of *Arabidopsis* in response to pathogens [22]. Plant stomata play an essential role in innate immunity against bacterial invasion; it has been discovered that flg22-mediated stomatal closure is conveyed by *Arabidopsis* lipoxygenase 1 (LOX1) and MPK3/6 [24]. Active oxygen species (AOS) formed in response to stimuli and throughout development can operate as signaling molecules in eukaryotes, leading to particular downstream reactions. It was shown that MPK6 and MPK3 can be triggered by varying concentrations of H_2_O_2_ in *Arabidopsis*. H_2_O_2_ also activates an oxidation-signal-induced type 1 (OXI1) kinase, which is a key component of ^1^O_2_ and H_2_O_2_ signaling and influences JA-dependent pathways and programmed cell death [29,30]. Exogenous H_2_O_2_ treatment of *Arabidopsis* protoplasts can greatly boost the activities of MPK3 and MPK6; however, in the *oxi1* mutant, the activity of MPK3/MPK6 was reduced when the protoplasts were treated with exogenous H_2_O_2_, and the resistance to downy mildew was decreased [29].

Due to the fact that simple nitrile formation upon tissue disruption depended almost entirely on NSP2 in seeds [31], it is necessary to reveal the function of NSP2 in the pathway of disease resistance. In this study, we revealed that NSP2 interacts with MPK3, and mutation or overexpression of *NSP2* results in a susceptible phenotype when challenged with *Pseudomonas syringae* pv. *tomato* DC3000 (*Pst* DC3000), demonstrating the necessity for *NSP2* to maintain a relatively stable expression level during disease resistance. Transcriptome and proteomic analysis revealed that NSP2 did respond to plant pathogen infection and plant hormone signal transduction. In addition, we predicted potential phosphorylation sites on NSP2 and confirmed its presence through observing phosphorylation modification of NSP2. Notably, after *Pst* DC3000 inoculation, the phosphorylation level of NSP2 was dramatically lowered, indicating a probable function for phosphorylation modification in NSP2 response to *Pst* DC3000. Furthermore, we found a substantial difference in MPK3 protein levels in *NSP2* overexpression lines before and after inoculation with *Pst* DC3000, indicating that MPK3 may play a role in NSP2 phosphorylation. Our finding of an NSP2 interacting protein showed a new role for NSP2 in plant disease resistance. Overall, our results emphasize the significance of NSP2 and its post-translational changes in plant-pathogen interactions.

## 2. Results

### 2.1. NSP2 Responds to Pst DC3000

Many studies suggest that NSP2 is involved in plant response to stress [8,31,32]. We explored whether *Pst* DC3000, a bacterial pathogen that causes strong plant defense responses, might regulate NSP2 expression. NSP2 expression indeed increased significantly in response to *Pst* DC3000 treatment, with the highest level of expression detected at 6 h post inoculation (Figure 1a), indicating that NSP2 is a component in the process of plant response to pathogen infection.

Next, we obtained T-DNA insertion mutant *nsp2* from Arabidopsis Biological Resources Center (ABRC) and *NSP2* overexpression line *NSP2*-OE by *Agrobacterium*-mediated transformation. To test whether the *NSP2* mutants have altered pathogen resistance, wild-type Col-0, *nsp2,* and *NSP2*-OE plants were inoculated with *Pst* DC3000. The total number of bacteria in *nsp2* and *NSP2*-OE leaves were both significantly higher than that in Col-0 leaves at 2 days post inoculation (dpi), suggesting that both *NSP2* lines exhibited inhibited resistance responses (Figure 1b).

Previous studies have shown that *NSP2* is primarily expressed in seeds [31]. In order to further investigate the tissue-specific expression pattern of *NSP2* in *Arabidopsis*, we analyzed the expression levels of *NSP2* in different plant tissues. Consistent with previous reports, our results demonstrated that *NSP2* was predominantly expressed in seeds, while it is also expressed in both roots and leaves (Figure 1c).

### 2.2. NSP2 Involved in the Plant Immune Signal Pathway at Transcriptional Level

To explore the disease resistance pathways involved in *NSP2*, we performed RNA sequencing (RNA-seq) analysis with wild type and *NSP2*-OE lines. Clean reads were mapped to *Arabidopsis* reference genome (TAIR10, www.arabidopsis.org); over 97% of the clean reads per library could be mapped to the reference database in the six individual libraries (Appendix A). These results indicated that the RNA-seq data were sufficient for subsequent gene expression analysis.

Then, a total of 234 differentially expressed genes (DEGs) between WT (Col-0) and *NSP2*-OE were identified with >1.5-fold differences in expression and *p*-value < 0.05. Among these, 139 genes were upregulated and 95 genes were downregulated in *NSP2*-OE (Figure 2a). To assess the correlation of gene expression patterns among the groups, systematic cluster analysis was performed and the heatmap result demonstrated the presence of numerous DEGs between WT and *NSP2*-OE (Figure 2b), indicating that the two materials marked disparities in gene expression patterns. 

The analysis of GO terms using the selected genes is useful in predicting the altered biological and molecular processes. Therefore, 234 DEGs were subjected to GO enrichment analysis to determine their functional significance in *NSP2*-OE. The significant enrichment was concentrated in several stress response biological processes, including iron ion starvation, cellular responses to hypoxia and decreased oxygen levels, as well as responses to chitin and aging (Figure 2c). KEGG pathway analysis suggested the DEGs were mainly involved in spliceosome, plant-pathogen interaction, and plant hormone signal transduction (Figure 2d). These results implied that *NSP2* might play an important role in the response process of plants to pathogens.

We tested the expression of four upregulated genes *RAB18*, *STMP2*, *STMP7*, *SAUR13* and four downregulated genes *MOP9*, *PDR7*, *SIKIC3*, *LRRAC1* to validate the RNA-seq results. All four upregulated gene transcripts were significantly higher in *NSP2*-OE while the four downregulated genes showed lower expression in *NSP2*-OE than in Col-0 (Figure 3a,b). Among them, S*TMP2* and *STMP7* are secretory transmembrane peptides unique to Brassicaceae, which function in plant growth and pathogen defense [33]. These results validated the RNA-seq data and confirmed that the NSP2 was related to plant-pathogen interaction.

### 2.3. NSP2 Affect Several Pathogen-Related Protein Levels by Proteomic Analysis

We further elucidated the mechanism of NSP2 involvement in disease resistance response using proteomic analysis. A total of 439 and 521 differential proteins (Log2 (fold change) | ≥ 1, *p* < 0.05) were identified in *NSP2*-OE and *nsp2* compared to Col-0, respectively. Among them, there were 206 upregulated and 233 downregulated proteins in *NSP2*-OE/Col-0, 247 upregulated and 274 downregulated proteins in *nsp2*/Col-0, respectively (Figure 4a). Further analysis found that four proteins were upregulated in *NSP2*-OE and downregulated in *nsp2* (AT1G10940, AT1G24280, AT2G29350, AT5G62440), while another four proteins were upregulated in *nsp2* and downregulated in *NSP2*-OE (AT2G03870, AT3G01520, AT5G04420, AT5G08060). It is worth noting that two of these eight genes, AT1G10940 (SnrK2.4) and AT2G29350 (SAG13), have been clarified to participate in the response of plants to stress and diseases [34,35]. These eight genes should be closely linked to NSP2 and used as valuable targets for in-depth research. 

GO enrichment analysis was carried out on the 439 and 521 differential proteins, respectively. The results showed that the differential proteins in *NSP2*-OE/Col-0 were mainly involved in the pathway of response to arsenic-containing substances, aging, and insect (Figure 4b). However, the differential proteins in *nsp2*/Col-0 are mostly engaged in the response to desiccation, jasmonic acid biosynthetic process, and programmed cell death at biological process portion (Figure 4c). These findings implied that NSP2 plays a wide range of and complicated roles in plant regulation, potentially mainly impacting stress responses and resistant hormone signaling pathways.

### 2.4. NSP2 Interacts with MPK3

In order to understand how NSP2 responds to *Pst* DC3000, NSP2 interacting proteins were detected by immunoprecipitation-tandem mass spectrometry analysis (IP-MS). MPK3 was co-precipitated with NSP2-GFP in NSP2 transgenic plants tagged with GFP and was chosen for further investigation since MAPK components are known to be involved in disease resistance.

The connection between NSP2 and MPK3 was established using a bimolecular fluorescence complementation (BiFC) assay. The vectors carried MPK3 fused with the YFP C-terminus and NSP2 fused with YFP N-terminus co-transform into *N. benthamiana* leaves. Bright fluorescence was observed in the nucleus and cytoplasm while no fluorescence was observed in the negative control. The findings validated NSP2’s interaction with MPK3 (Figure 5a). Co-immunoprecipitation (CoIP) assays obtained comparable results. Following the temporary co-expression of NSP2-MYC with MPK3-GFP in *N. benthamiana*, and the protein extract immunoprecipitation with anti-GFP antibodies, the band of NSP2-MYC was detected at the right position by Western blots (Figure 5b).

### 2.5. NSP2 and MPK3 Worked Synergistically in the Process of Disease Resistance

To study the role of NSP2 and MPK3 in regulating plant immunity, we generated the double mutant *nsp2mpk3* by crossing T-DNA insertion mutants of *NSP2* and *MPK3*. Phenotypic analysis revealed that the *nsp2mpk3* double mutant displayed similar phenotypes to the *nsp2* single mutant, both slightly smaller than the wild type (Figure 6a), suggesting that *NSP2* may be a function downstream of *MPK3*. The bacterial growth with *Pst* DC3000 inoculation of *nsp2mpk3* double mutant was higher than that in Col-0, which was similar to *mpk3* single mutant (Figure 6b). After *Pst* DC3000 inoculation, the expression level of disease-resistant related genes showed that *PR1*, *PR2,* and *PR5* were induced to varying degrees in each line and the expression levels of each gene in *nsp2*, *mpk3*, *nsp2mpk3,* and *NSP2*-OE were lower than that in Col-0 (Figure 6c–e). These results indicated that *NSP2* and *MPK3* share a pathway and synergistically promote disease resistance in plant.

Since reactive oxygen species (ROS) are key signal molecules that allow cells to respond quickly to varied stimuli and play an important role in plant defense responses, we performed DAB and NBT staining on Col-0, *nsp2*, *NSP2*-OE, *mpk3,* and *nsp2mpk3* leaves after inoculation with *Pst* DC3000. The staining area of DAB and NBT in *nsp2*, *NSP2*-OE, *mpk3,* and *nsp2mpk3* leaves after invasion of *Pst* DC3000 were significantly reduced compared with Col-0 (Figure 7a–c), indicating that the outbreak of ROS after invasion of *Pst* DC3000 was reduced compared with that of Col-0, which was consistent with the susceptible phenotype observed following *Pst* DC3000 inoculation.

Aniline blue staining was also used to detect callose deposition in leaves of Col-0, *nsp2*, *NSP2*-OE, *mpk3,* and *nsp2mpk3* after *Pst* DC3000 inoculation (Figure 7d). A large number of fluorescence signals could be detected in wild type; however, the fluorescence signals of *nsp2*, *NSP2*-OE, *mpk3,* and *nsp2mpk3* were very few (Figure 7e), indicating that the callose content of *nsp2*, *NSP2*-OE, *mpk3,* and *nsp2mpk3* was lower than that of Col-0, which was consistent with the fact that *nsp2*, *NSP2*-OE, *mpk3,* and *nsp2mpk3* were sensitive to *Pst* DC3000.

### 2.6. Pathogenic Inoculation Alters the NSP2 Phosphorylation Status 

We discovered that *Pst* DC3000 induces NSP2 transcriptional expression (Figure 1a), and that NSP2 protein content is likewise enhanced after pathogen treatment (Figure 8a). We used anti-MPK3 antibodies in above ground portions of *nsp2* mutant and NSP2-OE before and after *Pst* DC3000 treatment to examine the influence of NSP2 on MPK3 abundance. Interestingly, we detected a large rise in MPK3 abundance in NSP2-OE at normal condition; however, following *Pst* DC3000 treatment, the increasing trend disappeared and became similar to the wild type (Figure 8b). MPK3 is a protein kinase that typically conveys signals via phosphorylation during signal transduction. As a result, we postulated that MPK3 could phosphorylate NSP2. Based on the amino acid sequences of NSP2, we predicted that several high confidence sites in NSP2 could probably be phosphorylated by MPK3 (Appendix A). Following that, we used phos-tag gel assay to look at the phosphorylation alteration of NSP2. Surprisingly, we observed that NSP2 had phosphorylation modification; however, it was drastically reduced following *Pst* DC3000 treatment (Figure 8c), indicating that pathogen challenge may have modified the phosphorylation state of NSP2. Our data suggest that pathogenic inoculation alters the NSP2 phosphorylation status which might be caused by MPK3.

## 3. Discussion

*Pst* DC3000 is one of the most common and dangerous plant pathogens. It may infect not only *Arabidopsis thaliana*, but also many commercial crops, such as tomato and *N. benthamiana*, resulting in plant leaf damage and massive economic losses. The interaction between *Pst* DC3000 with the model plant *Arabidopsis thaliana* has become a valuable tool for studying the molecular mechanisms of plant disease resistance [36]. We discovered that *Pst* DC3000 induced *NSP2* expression, and that lack or overexpression of *NSP2* in plants resulted in higher sensitivity to *Pst* DC3000 (Figure 1a,b and Figure 6b), emphasizing the necessity of maintaining constant levels of *NSP2* during pathogen infection. The importance of protein homeostasis in disease resistance has been previously reported. Both overexpression and suppression of *OsLSR* lead to the autoactivation of immune response in rice [37], indicating that the transcript level of some key genes is strictly regulated by the immune system in plants.

Through transcriptome and proteomic analysis, we discovered that NSP2 participates in mRNA splicing, senescence, pathogen interaction, and hormone signal transduction (Figure 4), indicating that NSP2 plays an important role in the control of plant growth, development, and immunological responses. Functional studies of NSP family genes in *Arabidopsis* are currently uncommon. According to bioinformatic prediction, stress-related cis-regulatory elements are commonly found in the promoter region of NSP family genes, and NSP5 can considerably upregulate its expression under drought stress [38]. The structural variations in NSP proteins, as well as their enzyme activity in catalyzing the breakdown of glucosinolates, show that NSP family genes operate differently [8]. As a result, more studies on disease resistance of other genes in the NSP family are required.

The MAPK cascade pathway has been extensively investigated, demonstrating its critical function in plant resistance and defense response [39,40,41]. The MAPK cascade route associated with plant disease resistance has a large number of members at various levels, and these members may have cross-effects with other MAPK cascade pathways. In this work, we highlight the significance of NSP2 in controlling plant disease resistance by identifying its interacting proteins, and we discovered that NSP2 connected with MPK3. The *nsp2mpk3* double mutant showed a developmental behavior comparable to *nsp2* (Figure 6a), suggesting that NSP2 is genetically downstream of MPK3. Following *Pst* DC3000 inoculation, the double mutant demonstrated sensitive symptoms (Figure 6b). These findings indicate that we discovered a novel component for the MPK3 function complex in the immune signal transduction.

MPK3’s main role is to influence the expression of downstream genes and the regulation of physiological metabolism by phosphorylating downstream target proteins, which in turn affects plant growth and development and stress adaption [15,42,43]. We predicted NSP2 phosphorylation sites and postulated that NSP2 may be phosphorylated by MPK3, affecting its biological activity. Phosphorylation of NSP2 was indeed detected using phos-tag gel, and the modification was dramatically decreased following *Pst* DC3000 treatment (Figure 8b). Although no direct evidence relating NSP2 phosphorylation to MPK3 was revealed, our study laid the groundwork for future research into the potential of NSP2 being a direct target of MPK3. Therefore, we propose this possible regulatory pathway: under normal conditions, MPK3 phosphorylates NSP2 and phosphorylated NSP2 function to resist the infection of pathogens; upon treatment with *Pst* DC3000, the protein level of MPK3 decreases in *NSP2*-OE, resulting in a decrease in the phosphorylation level of NSP2, thus showing a susceptible phenotype; however, the cause of the reduction in MPK3 protein is uncertain. Of course, there is another possibility: after challenge by the pathogen, another unknown dephosphorylase was activated to interact with NSP2, resulting in a decrease in NSP2 protein phosphorylation. 

Overall, our study identifies a new role for NSP2 in pathogen resistance, sheds light on the function of NSP2 in plant-pathogen interactions, and establishes a connection between NSP2 and MPK3. Our result advances the understanding of plant defense mechanisms and provides a foundation for future research on its role in disease resistance modulation in *Arabidopsis*.

## 4. Materials and Methods

### 4.1. Plant Material and Growth Conditions

Genotypes used in this study were Col-0, *nsp2* (SALK_057194C), and *mpk3* (SALK_100651), the mutants were ordered from Arabidopsis Biological Resources Center (ABRC). Double mutant was created by genetic crosses, and F2 plants were genotyped for homozygotes. Transgenic plants were generated by the floral dip method. Surface-sterilized seeds were plated onto half-strength Murashige and Skoog medium (1/2 MS) for 7 days and then moved to soil and cultured in a growth chamber for 4–5 weeks (23 °C, 16/8 h light/dark). The illumination intensity was 90 μmol/m^2^/s.

### 4.2. Bacterial Cultivation and Inoculation

*Pseudomonas syringae* pv. *tomato* DC3000 (*Pst* DC3000) was grown as previously described [44]. Bacterial cells were rinsed with water, diluted in 10 mM MgCl_2_ (OD_600_ = 0.002), then infiltrated into *Arabidopsis* leaves. Thereafter, 4-week-old *Arabidopsis* plants grown in soil were sprayed with *Pst* DC3000 (OD_600_ = 0.2) or H_2_O (mock) for 24 h to examine resistance-related gene expression and reactive oxygen species accumulation. The samples were collected individually and kept at −80 °C until needed.

### 4.3. Reactive Oxygen Species Accumulation Determination

Reactive oxygen species (ROS) accumulation was measured by DAB (3,3′-diaminobenzidine) and NBT staining as previously described [45]. The leaves were darkened and kept at room temperature overnight after being placed in the staining solution (NBT dissolved in 25 mM KH_2_PO_4_ (pH = 7.6); the final concentration was 0.1 g/L). The staining solution was discarded, and the leaves were decolorized by boiling in the fixing solution (lactic acid: glycerol: ethanol = 1:1:3). The leaves were then placed in the new fixing solution, and images were taken after they had cooled.

### 4.4. Plasmid Construction

The full-length *NSP2* cDNA was amplified by RT-PCR and ligated to pENTR/D-topo vector (Thermo Fisher Scientific, K240020, Waltham, MA, USA). The pENTR-NSP2 plasmid was transferred into the binary vector pMDC83 to produce *p35S::NSP2-GFP* by LR recombination (Thermo Fisher Scientific, 11791-100). To generate the cCFP-MPK3 and nVenus-NSP2 fusion vector, the *MPK3* and *NSP2* cDNA were first cloned into the pSAT4-cCFP-C and pSAT1-nVenus-C vector, respectively. The DNA fragment containing cCFP-MPK3 and nVenus-NSP2 was released by I-SceI and AscI restriction enzyme, respectively and subsequently cloned into the pPZP-RCS2-ocs-bar vector [46]. All the primers are listed in Appendix A.

### 4.5. Bimolecular Fluorescence Complementation Assay

BiFC assays were performed as described by Lee et al. [46]. Paired cCFP and nVenus constructs were co-infiltrated in *N. benthamiana* leaves for 48 h. YFP signals were then detected with a confocal microscopy LSM 880 CLSM (Zeiss, Jena, Germany) at 488 nm.

### 4.6. Quantitative Real-Time PCR

The total RNA was extracted from the frozen *Arabidopsis* leaf samples with RNAiso Plus (Takara 9101). The resulting RNA samples were treated with DNase I (TransGen Biotech GD201-01) followed by reverse transcription using HiScript II Reverse Transcriptase (Vazyme R223-01) according to the manufacturer’s instructions. qRT-PCR was performed in triplicate using SYBR Green Real-Time PCR Master Mix (Vazyme Q711-02) on the Bio-Rad CFX96 system. The 2^− ∆∆ Ct^ method was used to calculate the relative gene expression level across the samples. Finally, the results were presented as histograms by GraphPad Prism 8 software (GraphPad, San Diego, CA, USA). Primers used are listed in Appendix A.

### 4.7. Western Blot Analyses

For extraction of total proteins, 4- to 5-week-old rosette leaves in soil were ground to a fine powder in liquid nitrogen, and homogenized in two volumes of lysis buffer (50 mM Tris-HCl (pH 8.0), 150 mM NaCl, 0.2% Nonidet P-40, 2 mM DTT, 10% glycerol, protease inhibitor) and centrifuged for 15 min at 12,000 rpm. The supernatant was mixed with 5 × SDS loading buffer and boiled at 95 °C for 5 min. After centrifugation at 12,000 rpm for 1 min, the supernatant containing total proteins was then separated on SDS-PAGE gels. Western blot analyses were carried out using anti-GFP (TransGen Biotech HT801, Beijing, China), anti-MYC (TransGen Biotech HT101), anti-Actin (CWBIO CW0264M), and anti-MPK3 (ABclonal A0228). Western blots were developed with ECL+ (Vazyme E412-01), detected with ChemiDoc XRS+, and quantified using the ImageJ software.

### 4.8. Co-Immunoprecipitation

Four-week-old whole plants of Arabidopsis in soil were collected. Total protein was extracted with the IP buffer containing 50 mM Tris-HCl, pH 8.0, 150 mM NaCl, 2% β-mercaptoethanol, 0.1 mM PMSF, 0.5% Triton X-100, 5% glycerol, and protease inhibitor cocktail (Roche). To test the interaction of NSP2 with MPK3, MPK3-GFP was co-expressed with NSP2-MYC in *N. benthamiana*. Protein extract was then incubated with Anti-GFP Nanobody Agarose Beads (KT Healthy KTSM1301). Anti-GFP and anti-MYC antibodies were used to detect MPK3-GFP and NSP2-MYC, respectively.

### 4.9. RNA Sequencing 

Total RNA was extracted from samples of different genotypes. Sequencing libraries were constructed by Novogene (Beijing, China), with PE150 mode sequencing being performed on the Illumina sequencer. Prior to sequence alignment, the obtained reads were subjected to trimming using Trimmomatic (version 0.39) [47], and then aligned to the reference genome of *Arabidopsis thaliana* (TAIR, version 10) using HISAT2 (version 2.2.1) [48]. Differential expression analysis of the two comparison combinations was performed using DESeq2 (version 1.40.1) [49] with a significant change being defined as the threshold (|log2 fold change| > 1.5; *p*-adjust < 0.05). GO enrichment and KEGG analysis of differentially expressed genes were performed using clusterProfiler (version 1.40.1) [50].

## Figures and Tables

**Figure 1 plants-12-02857-f001:**
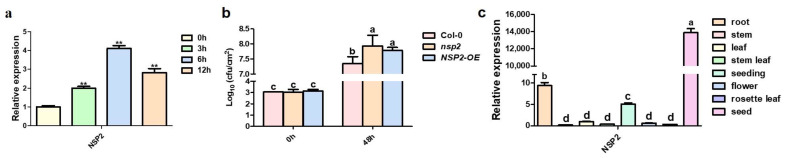
*NSP2* responds to *Pst* DC3000. (**a**) *NSP2* expression after *Pst* DC3000 infection. Four-week-old Col-0 were sprayed with suspension of *Pst* DC3000 (OD_600_ = 0.2 in 10 mM MgCl_2_) on leaves. Expression of *NSP2* was detected by qRT-PCR at 0 h, 3 h, 6 h, and 12 h post *Pst* DC3000 treatment, respectively. *Actin2* was used as an internal control. Transcript levels in Col-0 were set as 1. Three independent reactions were performed for each technical replicate. Two technical replicates were performed for each biological replicate. At least two biological replicates showed similar results. Error bars represent standard deviation and ** indicates *p* < 0.01 (paired student’s *t*-test). (**b**) Measurement of bacterial titers of *Pst* DC3000. Col-0, *nsp2* mutants, and *NSP2*-OE lines were infiltrated with a suspension of *Pst* DC3000 (OD_600_ = 0.002 in 10 mM MgCl_2_). Bacterial growth was determined at 0 h and 48 h. Each data point consisted of at least four samples. Experiments were repeated three times. Data represent the means of three replicates ± SDs (error bars). Different lowercase letters indicate a significant difference between different lines. Statistically significant differences were determined by one-way ANOVA and *p* < 0.05. (**c**) Tissue expression pattern of *NSP2*. Tissues of different periods were extracted from Col-0 for RNA extraction. Expression of *NSP2* was detected by qRT-PCR, and *Actin2* was used as an internal control. Three independent reactions were performed for each technical replicate. Two technical replicates were performed for each biological replicate. Error bars represent standard deviation. Different lowercase letters indicate a significant difference between different lines.

**Figure 2 plants-12-02857-f002:**
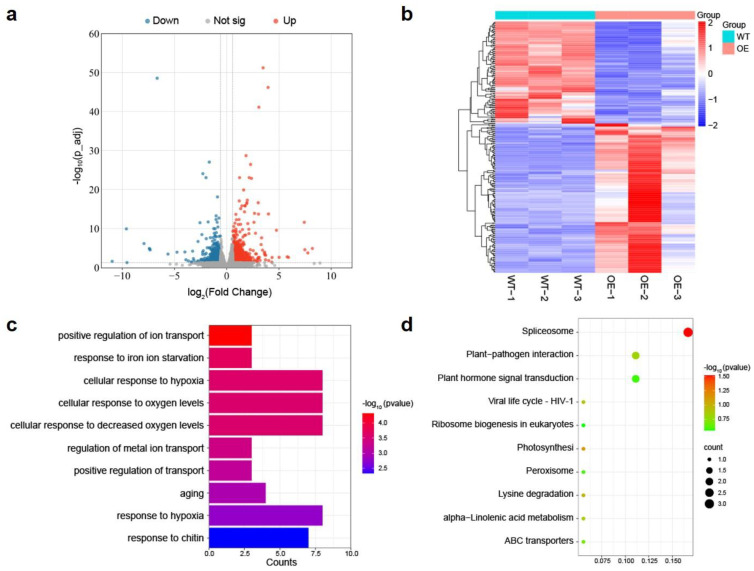
Transcriptome analysis of WT (Col-0) and *NSP2*-OE. (**a**) RNA-seq data visualized by volcano plots. Each point represents a gene. Red points represent upregulated genes, and blue points represent downregulated genes. | Log2 (fold change) | ≥ 1.5 or FDR-adjusted *p*-value < 0.05 are marked with black dashed lines. (**b**) Hierarchical clustering and heatmap of genes that vary in expression between WT and *NSP2*-OE. In comparison to the control cohort, the shown DEG exhibited log2 FC ≥ 1.5fold increase in gene expression intensity and the FDR-adjusted *p*-value < 0.05. (**c**) Enriched GO terms in DEGs between WT and *NSP2*-OE. The top ten GO terms were associated with the coding gene function. (**d**) Enriched KEGG pathways in DEGs between WT and *NSP2*-OE. The top ten KEGG pathways were associated with the coding gene function.

**Figure 3 plants-12-02857-f003:**
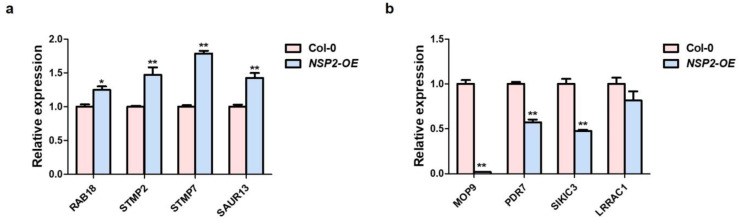
Verification of upregulated and downregulated genes in *NSP2*-OE lines. The qRT-PCR analysis of *RAB18*, *STMP2*, *STMP7*, *SAUR13* (**a**) and *MOP9*, *PDR7*, *SIKIC3*, *LRRAC1* (**b**) in 4-week-old Col-0 and *NSP2*-OE lines. *Actin2* was used as an internal control. Transcript levels in Col-0 were set as 1. Three independent reactions were performed for each technical replicate. Two technical replicates were performed for each biological replicate. Error bars represent standard deviation and * indicates *p* < 0.05; ** indicates *p* < 0.01 (paired student’s *t*-test).

**Figure 4 plants-12-02857-f004:**
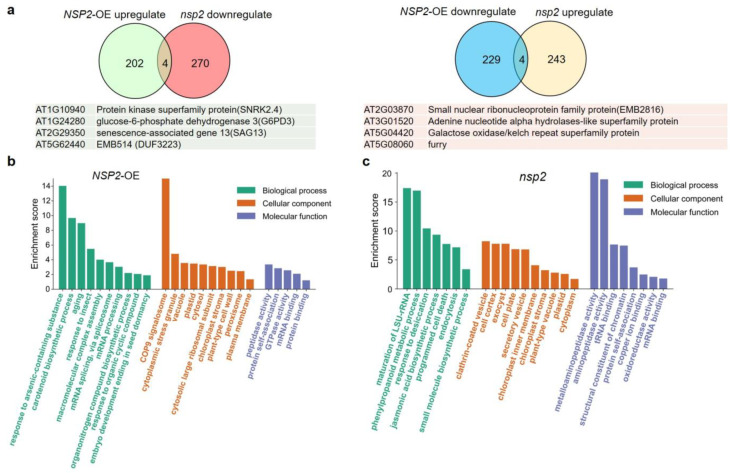
Proteomic analysis of NSP2. (**a**) Venn diagram compares the differentially expressed proteins in *NSP2*-OE/Col-0 and *nsp2*/Col-0. | Log2 (fold change) | ≥ 1 and FDR-adjusted *p*-value < 0.05, relative to the control cohort. (**b**) Enriched GO terms in differential proteins between *NSP2*-OE and Col-0. (**c**) Enriched GO terms in differential proteins between *nsp2* and Col-0.

**Figure 5 plants-12-02857-f005:**
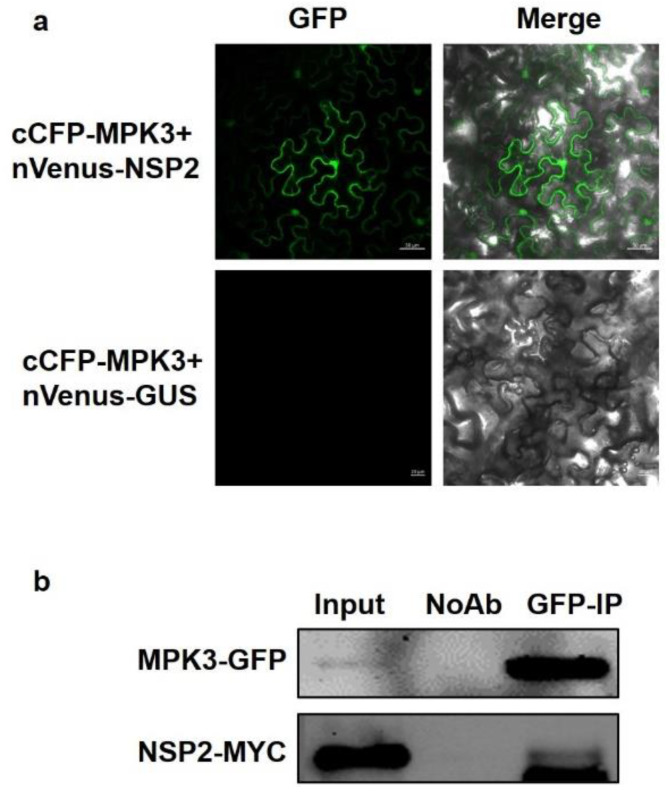
MPK3 interacts with NSP2. (**a**) BiFC analysis of NSP2 with MPK3. Paired cCFP-MPK3 and nVenus-NSP2 fusion proteins were co-infiltrated into *N. benthamiana* leaves. The BiFC signal (yellow fluorescence) was detected at 48–72 h after infiltration by confocal microscopy, assigned as green color. Thirty cells were examined and an image is shown. (**b**) Co-immunoprecipitation between NSP2 and MPK3. NSP2-MYC was co-expressed with MPK3-GFP in *N. benthamiana* leaves. Anti-GFP and anti-MYC antibodies were used to detect GFP- and MYC-fused proteins, respectively.

**Figure 6 plants-12-02857-f006:**
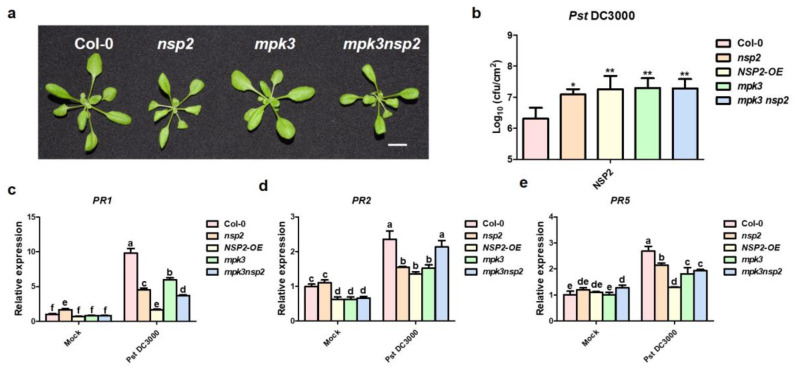
NSP2 and MPK3 worked synergistically in the process of disease resistance. (**a**) Vegetative phenotype of *nsp2*, *mpk3,* and *mpk3nsp2*. Aerial parts of 4-week-old plants were photographed (scale bar, 1 cm). (**b**) Measurement of bacterial titers of *Pst* DC3000. Col-0, *nsp2*, *mpk3*, *mpk3nsp2,* and *NSP2*-OE lines were infiltrated with *Pst* DC3000. Bacterial growth was determined at 48 h. Each data point consisted of at least four samples. Experiments were repeated three times. Data represent the means of three replicates ± SDs (error bars). Statistically significant differences were determined by one-way ANOVA and * indicates *p* < 0.05; ** indicates *p* < 0.01. (**c**–**e**) Expression of *PR1*, *PR2,* and *PR5* after *Pst* DC3000 infection. The 4-week-old Arabidopsis were sprayed with H_2_O (mock) and *Pst* DC3000 (OD_600_ = 0.2). Expression of *PR1*, *PR2,* and *PR5* was detected by qRT-PCR at 24 h post H_2_O and *Pst* DC3000 treatment, respectively. *Actin2* was used as an internal control. Transcript levels in Col-0 treated with H_2_O were set as 1. Three independent reactions were performed for each technical replicate. Two technical replicates were performed for each biological replicate. Error bars represent standard deviation. Different lowercase letters indicate significant differences between different lines. Statistically significant differences were determined by one-way ANOVA and *p*-values were < 0.05.

**Figure 7 plants-12-02857-f007:**
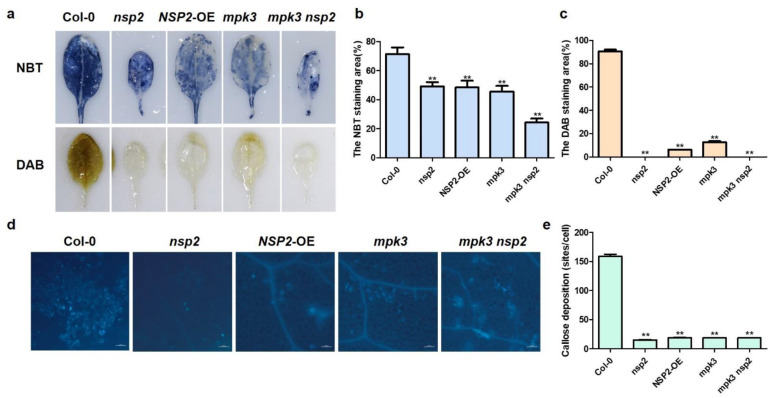
The accumulation of callose and reactive oxygen species (ROS) after inoculation with *Pst* DC3000. (**a**) The leaves of Col-0, *nsp2*, *mpk3*, *mpk3nsp2,* and *NSP2*-OE inoculated with *Pst* DC3000 were cut and placed in the NBT or DAB staining solution, treated in the dark at room temperature overnight, and photographed after decolorization. (**b**,**c**) Quantification of the NBT and DAB staining area in the leaves was statistically analyzed by the ImageJ software. Error bars represent standard deviation. Statistically significant differences were determined by one-way ANOVA and ** indicates *p*-values were < 0.01. (**d**) The leaves of Col-0, *nsp2*, *mpk3*, *mpk3nsp2,* and *NSP2*-OE inoculated with *Pst* DC3000 were cut and soaked in 95% alcohol. Until the leaves fade and become transparent, they were transferred to the dye solution (150 mmol/L K_2_HPO_4_, pH = 9.5, 0.01% aniline blue) for 30 min, washed with sterile water, and then observed under a slide fluorescence microscope. (**e**) Quantification of callose in the visual field area was statistically analyzed by the ImageJ software. Error bars represent standard deviation. Statistically significant differences were determined by one-way ANOVA and ** indicates *p*-values were < 0.01.

**Figure 8 plants-12-02857-f008:**
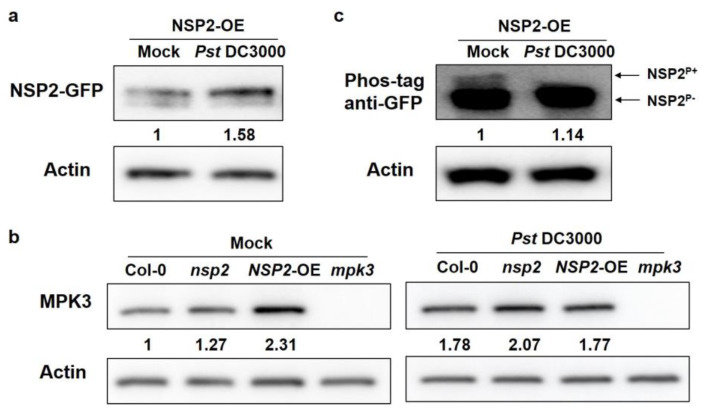
Western blot analysis of *nsp2*, *NSP2*-OE, and *mpk3* plants. (**a**) NSP2 was induced after *Pst* DC3000 treatment. Protein extracts were prepared from the 4−week−old *NSP2*-OE lines sprayed with H_2_O (mock) and *Pst* DC3000 (OD_600_ = 0.2). Western blot was probed with anti-GFP antibody, ACTIN was used as an internal reference of the total protein. (**b**) *Pst* DC3000 affects the phosphorylation of NSP2. Proteins were separated in a phos−tag gel, and NSP2 was detected with anti-GFP antibody. ACTIN was used as an internal reference of the total protein. (**c**) Abundance of MPK3 in Col-0, *nsp2*, *NSP2*-OE, and *mpk3* plants. Protein extracts were prepared from Col-0, *nsp2*, *NSP2*-OE, and *mpk3* plants sprayed with H_2_O (mock) and *Pst* DC3000 (OD_600_ = 0.2). Western blot was probed with anti-MPK3 antibody, ACTIN was used as an internal reference of the total protein. The protein contents were quantified by the ImageJ software.

## Data Availability

The RNA-Seq and proteome data used in this study are available from the appropriate authors upon reasonable request.

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
