# Peer review of "Nitrile-Specific Protein NSP2 and Its Interacting Protein MPK3 Synergistically Regulate Plant Disease Resistance in *Arabidopsis"

_plants, 2023, doi:10.3390/plants12152857_

Round 1
Reviewer 1 Report
Dear Authors,
The manuscript Zhai et al. is a nicely built work on NSP. All the experiments and analysis has been well planned.
I have few suggestions which can be incorporated.
1. Stats for figure 1c
2. A quantification for DAB and NBT staining would be very useful to show the significance of these two genes.
3. Please mention the method used for real time quantitative analysis in the material methods section.
4. Through the paper synergy between NSP2 and MPK3 has been described. Can the authors discuss the probable pathway for this in the discussion? Probably a model would be very useful in the conclusion. You have the phosphorylation data well acquired. Use it to decipher a hypothetica model.
5. Do you have any quantification of the NSP2 and MPK3 expression in the RNA seq analysis? If yes, please add it in the supplementary.
Reviewer 2 Report
The article "Nitrile-specific protein NSP2 and its interacting protein MPK3 2 synergistically regulate plant disease resistance in Arabidopsis" is a well-organized study. The execution and representation of the study are sound. The article enlightens the new mode of synergistic action between NSP2 and MPK3 in the disease resistance process.
Some minor changes are required:
The abstract needs to be more elaborated and take home message is missing which should be added at the end of the abstract. In the introduction section, the authors should explain the motive of the study. The materials and methods and results section is acceptable, but the discussion is fragmented and must link the whole story with references. Avoid presenting results in the discussion.
Minor grammatical errors must be addressed.
